# Morphological Evolution of Passive Soil Arch in Front of Horizontal Piles in Three Dimensions

**Xiang Ren [1], Lijuan Luo [1,2,*], Yunxin Zheng [1], Jiakuan Ma [1] and Xuexu An [1]**

[1]  School of Civil Engineering, Chang'an University, Xi'an 710061, China; renxiang@chd.edu.cn (X.R.); 2020002800@chd.edu.cn (Y.Z.); mj815663129@163.com (J.M.); 2019028012@chd.edu.cn (X.A.)
[2]  Institute of Underground Structure and Engineering, Chang'an University, Xi'an 710061, China
*   Correspondence: luojuan@chd.edu.cn; Tel.: +86-029-8233-7356

**Abstract:** The anti-slide pile is a primary method of landslide control. The effect of the passive soil arch in front of the embedded section of piles has a significant effect on the anti-slide pile's bearing capacity. The upgraded model test scheme was used to conduct model tests with a pile spacing four times the width of the pile and a geometric scale ratio of 1:15. The anti-slide pile stress, pile bending strain, and soil stress in front of the pile were all studied in relation to the loading amount. In addition to the model test, the numerical simulation method was utilized to investigate the three-dimensional morphological change of the passive soil arch in front of the pile. The results indicated that: clearly, the side piles can eliminate the border effect. The distribution of pile bending strain along the pile after loading is referred to as a parabola. Bending failure occurred at a depth of 40 mm, approximately 0.9 m from the pile top. Under the condition that the pile spacing is four times the pile width, a passive soil arch occurs in front of the anti-slide pile's fixed part, and its development can be split into four stages: formation, development, completion, and destruction. The passive soil arches in front of the piles are generated and destroyed gradually along the buried depth, and the three-dimensional surface of the space drops gradually along the buried depth with the loading amount and advances toward the loading direction until the anti-slide pile system fails. The research findings and experiences can serve as a basis for future research.

**Keywords:** anti-slide pile; embedded section; passive earth arch; model test; spatial form

## 1. Introduction

The "soil arch effect" is a regular occurrence in geotechnical engineering. The creation of soil arches is mostly owing to the "wedge tightening" impact of uneven displacement on soil particles, resulting in the soil arching effect [1–3]. Following the turn of the twenty-first century, the application of the soil arch effect to the thrust research of landslides behind anti-slide piles has garnered widespread attention. Theoretical analysis, model testing, and numerical simulation have all been used to conduct fruitful research. Researchers investigated the creation, development, and collapse mechanisms of soil arches behind and between piles. The relationship between the soil arch effect between piles and the anti-slide pile design parameters is discussed. Numerous significant accomplishments have been made. Wang created a model for calculating pile spacing based on the static equilibrium condition of soil arches between piles [4]. Following academics have produced suitable adjustments to the aforementioned models from a variety of angles, including slope inclination angle [5], constitutive model [6,7], and balance condition of soil arch stress [8–10]. Generally, scholars assumed that the soil arch's geometry is parabolic (reasonable arch axis). The thrust force generated by a landslide behind a pile is seen as a weight that is spread uniformly. The link between thrust force and pile spacing (clear distance) is determined using the static equilibrium condition at the arch foot and the soil strength criterion at the arch foot and vault, and the appropriate pile spacing is determined using known or available landslide thrust.

Existing research on horizontal soil arch can be classified into three categories:

(1) Direct observation, as demonstrated by Jiang [11] and Dou [12], or scanning and observing the interior of soil using novel methods and technologies, as demonstrated by Jin [13] who used infrared imaging technology to study the soil arching effect from an energy perspective and Chen [14] who used transparent soil technology. This strategy is only appropriate for qualitative analysis; it is not appropriate for reasonable quantitative research.

(2) A systematic research of the soil arching effect was conducted by combining model testing with theoretical analysis or numerical simulation [15–20]. This is the primary approach of research at the moment.

(3) Soil stress was measured during pile-soil contact by embedding a soil pressure sensor, and the soil arch effect was investigated using the soil stress distribution law, in order to deduce the characteristics of soil arch [21–23].

Current research on the soil arch effect focuses primarily on the expansion of the soil arch behind the cantilever section of anti-slide piles, highlighting that the soil arch effect is a manifestation of pile-soil interaction and that its characteristics between two piles can be regarded as a manifestation of the pile group effect, which is primarily used to determine the maximum or optimal pile spacing under a specific sliding force. According to the formation mechanism and stress characteristics of the soil arch, the passive soil arch effect in front of the pile should exist and have a substantial impact on the bearing capacity. The current study on the passive soil arch effect in front of piles generated by the interaction between horizontally strained piles (such as an embedded anti-slide pile) and soil in front of piles is insufficient [21,22]. Based on prior research on soil arch in front of piles, this article enhances the test scheme in light of the shortcomings of the typical horizontal stress pile model test. We tested a passive soil arch in front of an embedded portion of anti-slide piles with a spacing four times the width of the pile. To identify the axis of the passive soil arch in front of piles, the contact stress in front of the pile, the bending moment of the pile body, and the soil stress in front of the pile are studied and fitted. To supplement the model test, a realistic three-dimensional numerical model was constructed, and the three-dimensional spatial distribution of passive soil arch in front of piles was investigated using the flexibility of the numerical simulation approach. The research results and experience will reveal the spatial form of passive soil arch in front of the piles, serve as a reference for future research on the bearing capacity of anti-slide piles taking passive soil arch into account, and enhance the theory of pile foundation calculation.

## 2. Experimentation

The test system is built primarily of three components: a model framework, a data collecting system, and a loading system (Figure 1). To assure the loading system's stability during the test, the existing model box was reinforced horizontally with H-section steel. The model box was 5 m in length, 2.5 m in width, and 2 m in height. Figure 2 depicts the model's schematic diagram.

In general, an obvious boundary effect appears in the model test scheme for horizontal stressed piles (Luo, 2015, 2017), which has a significant effect on the stress distribution of the soil in front of the pile and is easily responsible for reducing the dependability of test data. As a result, this research optimizes the present standard test scheme: This test's primary study objective is to evaluate three anti-slide piles and the change rule for soil stress in their vicinity. Five model piles are established, along with one side pile on each side of the primary research area (as shown in Figure 1).

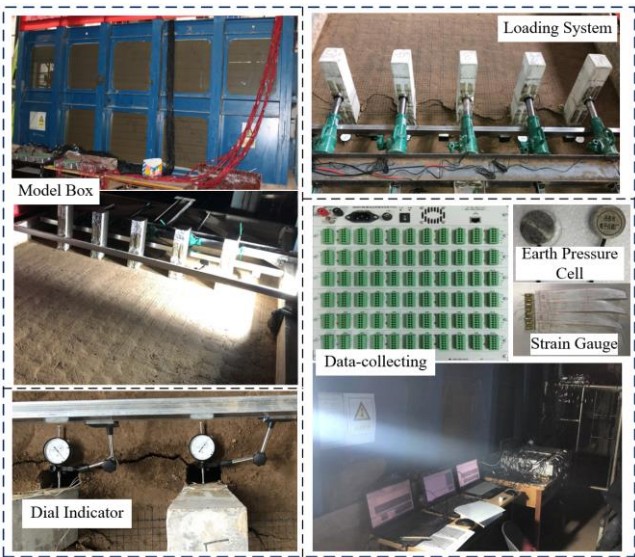

**Figure 1.** The model test system.

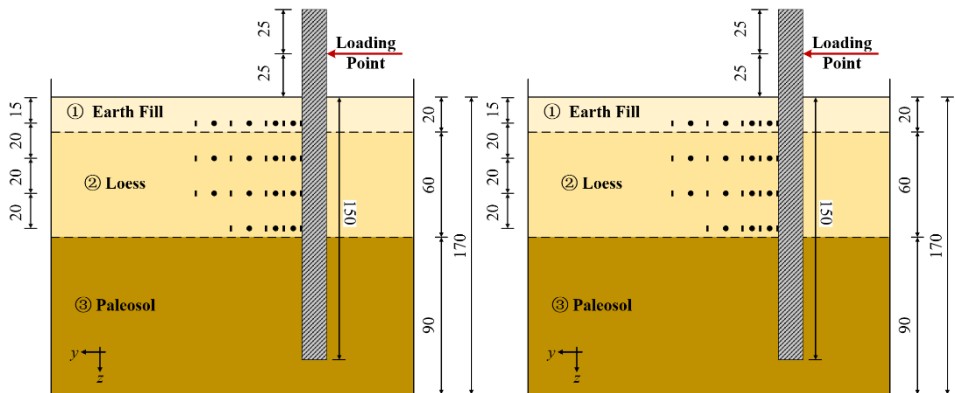

**Figure 2.** Diagram of the model (unit: m).

## 2.1. Model Similarity Ratio

This test is mostly for the purpose of determining the changing law of soil stress in front of a pile, and the similarity between the soil and pile elastic modulus must be established first. Dimensional analysis is used to determine the similarity ratios of major physical quantities, and the specific similarity constants are listed in Table 1.

**Table 1.** Similarity constants of main physical quantity.

| Physical Quantities | Similarity Constant |
| --- | --- |
| Geometry | $C_l = 15$ |
| Modulus of elasticity | $C_E = 2$ |
| Strain | $C_\varepsilon = 1$ |
| Stress | $C_\sigma = C_E\, C_\varepsilon = 2$ |
| Poisson ratio | $C_\mu = 1$ |
| Concentrated load | $C_F = C_\sigma\, C_l^2 = 450$ |
| Linear load | $C_q = C_\sigma\, C_l = 30$ |
| Area load | $C_p = C_\sigma = 2$ |
| Moment | $C_M = C_\sigma\, C_l^3 = 6750$ |

### 2.2. Model Materials

2.2.1. Model Piles

In this model test, anti-slide piles on a loess slope are used as the prototype, with the prototype pile embedded to a depth of 20 m. The pile body measures 1.5 m × 2 m in section. Pouring concrete with a compressive strength of 30 MPa and an elastic modulus of 31.5 GPa is C30.

The length of the model pile is 2.0 m, the section size is $a \times b = 14$ cm × 10 cm (Figure 3), the lengths of the cantilever and fixed segments are 50 cm and 150 cm, respectively, and the pile spacing is $L = 4b = 40$ cm, and the layout scheme is illustrated in Figure 4.

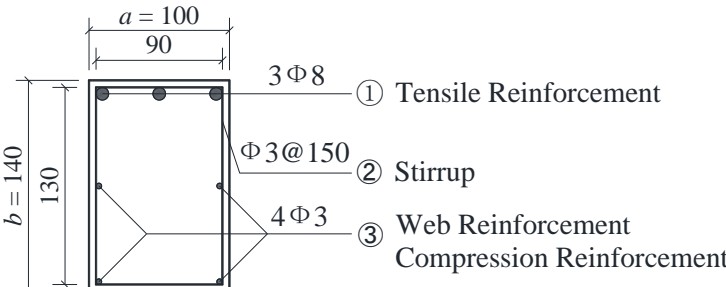

**Figure 3.** Structural reinforcement diagram of model pile (unit: mm).

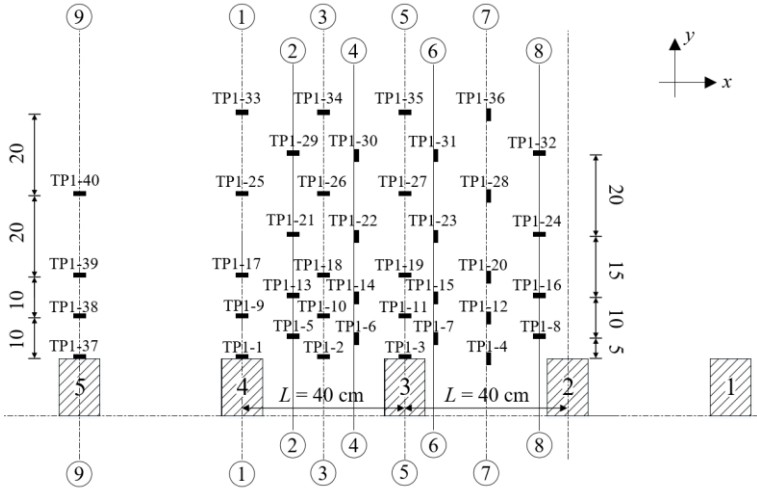

- The numbers in bars indicate earth pressure chamber numbers (e.g. TP1-1);
- The numbers in circles indicate the numbers of axes (e.g. ①-①).

**Figure 4.** Plane layout of earth pressure chamber of the first floor (unit: cm).

The concrete used in the model piles has a strength grade of C15 ($f_{cu,k} = 15$ MPa). As shown in Table 2, the mixture ratio was calculated by a large number of experiments, and the compressive strength is 16.59 MPa. The reinforcement ratio of a concrete structure is chosen in accordance with the principle of equal strength (Equation (1)), as illustrated in Figure 3. Reinforcement has a yield strength of 235 MPa and a maximum strength of 310 MPa.

$$\frac{A_{ps}f_{py,k}}{A_{pc}f_{pcu,k}} = \frac{A_{ms}f_{my,k}}{A_{mc}f_{mcu,k}} \tag{1}$$

where $A_{ps}$ is the reinforcement area of the prototype structure. $A_{pc}$ is the concrete area of the prototype structure. $f_{py,k}$ is the standard value of tensile strength of the reinforced bars of the prototype structure. $f_{pcu,k}$ is the standard value of compressive strength of concrete cube of the prototype structure. $A_{ms}$ is the reinforcement area of the model structure. $A_{mc}$ is the concrete area of the model structure. $f_{my,k}$ is the standard value of tensile strength of

reinforced bars of model structure. $F_{\text{mcu,k}}$ is the standard value of compressive strength of concrete cube of model structure.

**Table 2.** Mixture ratio of model pile concrete.

| Cement | Water | Fly-Ash | River Sand | Coarse Aggregate |
|--------|-------|---------|------------|------------------|
| 1 | 0.93 | 0.35 | 4.23 | 4.43 |

2.2.2. Model Soil

The model soil was created by crushing and screening the original loess and layering it, with the compression modulus serving as the primary controlling parameter. Compaction degree (compaction number) tests are used to evaluate the compression modulus of each layer during the filling process. Considering the real operating state and bearing mode of anti-slide piles, the model test does not account for the gap between the soil and the pile, and pile-soil contact is deemed to be satisfactory. Following loading, soil samples were collected at intervals of 10, 40, 70, 100, 130, and 160 cm to conduct additional geotechnical testing. After comparison and validation with the data collected during the filling process, the model soil's physical and mechanical indexes were computed, as shown in Table 3, and the profile of the model soil layer is illustrated in Figure 2.

**Table 3.** Physical and mechanical index of model soil.

| Soil Layers | Volumetric Weight (kN·m$^{-3}$) | Void Ratio | Cohesive Strength (kPa) | Internal Friction Angle (°) | Compression Modulus/MPa | | |
|-------------|---------------------------------|------------|-------------------------|-----------------------------|-------------------------|---------------|-----------------|
| | | | | | Model Soil | Prototype Soil | Similarity Ratio |
| ① Earth fill | 17.87 | 0.76 | 35.45 | 28.15 | 10.45 | 20 | 1.91 |
| ② Loess | 17.13 | 0.84 | 27.20 | 29.95 | 8.10 | 16.8 | 2.07 |
| ③ Paleosol | 18.38 | 0.72 | 34.50 | 28.4 | 10.85 | 26 | 2.39 |

*2.3. Sensors Arrangement*

2.3.1. Layout of Earth Pressure Cells

Existing research on the soil arch effect indicates that the axis of the soil arch is the reasonable axis of a three-hinged arch, and the direction of major stress at any location on the arch axis is tangent to that point, resulting in equal horizontal stress components at each point. As a result, the axis of the soil arch should correspond to the horizontal stress ($\sigma_x$) contour in front of the pile.

Four layers of earth pressure boxes are installed in front of the pile in accordance with the passive earth arch's characteristics. Each layer of the earth pressure box follows the same layout and naming conventions. The earth pressure distribution method is designed on the following principle: using model symmetry, the $\sigma_x$ and $\sigma_y$ of measuring places in the range between two piles of three model piles in the middle are determined. Finally, the findings are aggregated into a single range between piles, and the stress values for each soil point are obtained in two horizontal directions for examination.

Taking the first layer as an example (Figure 4), the earth pressure cells are distributed differently on both sides of axis ⑤ (symmetric axis). Earth pressure cells of Type A measure earth pressure in the y direction (along axis ①), while earth pressure cells of Type B measure earth pressure in the x direction (along axis ④). The central axes of piles are ①, ⑤, and ⑨, while the central axes of adjacent heaps are ③ and ⑦. Axes ② and ④ are the quarter lines of pile net distance, while axes ⑥ and ⑧ are positioned at the pile edge's 2 cm inward offset. Four layers of earth pressure boxes are laid vertically (Figure 2) at buried depths of 25 cm, 35 cm, 55 cm, and 75 cm. The fourth layer of earth pressure cells has a greater buried depth and a lower pile-soil mutual displacement, allowing for the elimination of two rows of earth pressure boxes.

The test utilized Bx-2 strain type earth pressure cells. It had a range of 0.6 MPa, a precision of 1/1000, and a diameter of 2 cm. The method of burying earth pressure cells is standardized in the test: after filling the soil to a depth of 5 cm above the design height, a hole of 6 cm is dug (considering the radius of the earth pressure cells), and then one earth pressure cell is buried; the backfill soil should be ground and backfilled in equal quantities to the excavated soil, to ensure that the backfill soil's density is close to that of the surrounding soil.

### 2.3.2. Layout of Strain Gauges

Strain gauges were installed at a 20 cm interval on all five model piles. For each pile, $2 \times 9 = 18$ pieces are placed in front (compression side) and behind (tension side), and the numbers are correspondingly S*F1~S*F18 in front of the pile and S*B1~S*B18 behind the pile (beginning from the pile top). '*' denotes the total number of anti-slide piles (as shown in Figure 5). In the original scheme, epoxy resin was employed as the protective coating of the strain gauge rather than dolomite to minimize the effect on pile stiffness.

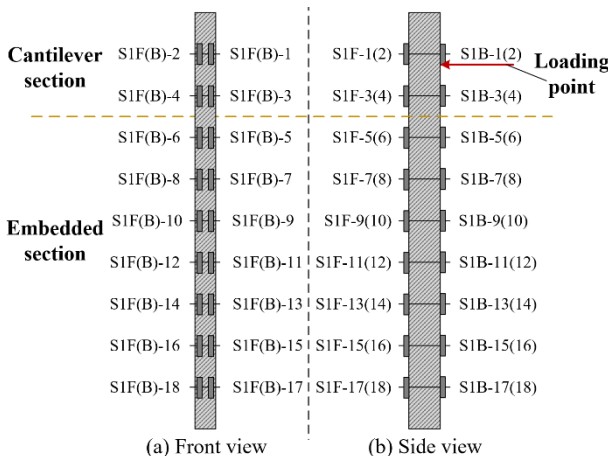

**Figure 5.** Arrangement of strain gauges of pile 1.

### 2.4. Loading and Data Collection Schemes

After burying the model pile, plastic film was placed on the soil surface to retain moisture, and the piles were filled three days later. A horizontal jack is installed 0.25 m from the top of each pile at the cantilever section for loading [24]. Between the jacks and the piles, force sensors were installed to gather thrust force throughout the loading process and to examine the development law of various pile forces. Dial indicators were placed behind the pile at the soil surface, and the displacement of the pile at the soil surface was used as the loading control variable, indicating. Five horizontal jacks were simultaneously and equally adjusted during loading to achieve steady and synchronous loading. After applying each amount of force, the jacks remained immobile until the data were steady. The DH3816 data collecting equipment was utilized to simultaneously collect data for horizontal thrust, pile strain gauge, and earth pressure in front of the pile. The following are the specific loading steps:

(1)　The 0.1 mm preload was used as the initial state of the test, 30 min after the data acquisition instrument's data balance;

(2)　Each stage was loaded with 0.5 mm, and data were collected after 30 min of static loading until $\delta = 10$ mm;

(3)　Each stage was loaded with 1 mm, and data were collected after 45 min until $\delta = 40$ mm;

(4)　Each stage was loaded with 2 mm, and data were collected after standing for 80 min until $\delta = 70$ mm.

### 3. Analysis of Test Results

When processing and evaluating data for horizontal thrust and bending strain of anti-slide piles, the data for piles 1 and 5 are averaged, which is referred to as side pile data. The data from piles 2 and 4 are averaged, resulting in what is referred to as secondary side pile data. Identify pile 3 as the central pile.

#### 3.1. Horizontal Thrust

The force sensors installed between the jacks and the piles are utilized to gather the corresponding jack thrust at various loading levels, and the thrust curve for each pile is plotted against the loading amount, as shown in Figure 6. As the loading quantity increases, the thrust of the pile by the jack can be divided into two stages: the bearing stage and the failure stage. The thrust grows progressively with the loading quantity and the growth rate reduces gradually during the bearing stage. When the loading quantity is increased to 40 mm, the thrust is essentially constant, while the side pile thrust drops, indicating that the anti-slide pile system has entered the failure stage.

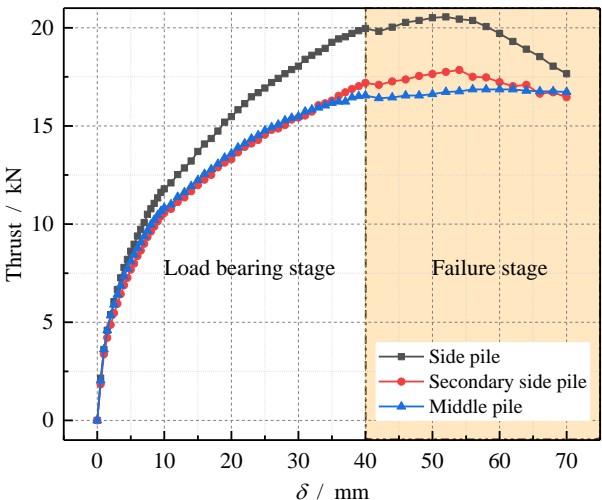

**Figure 6.** Horizontal thrust of the jacks.

In the bearing stage, the thrust values of the secondary side pile and the middle side pile are quite similar, and can be considered identical and smaller than the side pile. This indicated that the presence of the side pile effectively eliminates the boundary effect problem in the model test system of five anti-slide piles, and that it is reasonable to investigate the passive soil arch effect in front of the embedded pile using the soil between the middle pile and the secondary side pile.

#### 3.2. Bending Moment of the Piles

Equation (2) illustrates the calculation formula for pile bending strain [25].

$$\varepsilon_{\mathrm{M}} = (\varepsilon_{+} - \varepsilon_{-})/2. \tag{2}$$

The following formula can be used to determine the bending moment of a pile.

$$M = EW\varepsilon_{\mathrm{M}} \tag{3}$$

where $\varepsilon_{\mathrm{M}}$ denotes the bending strain of the pile body, $\varepsilon_{+}$ denotes the strain behind the pile, $\varepsilon_{-}$ denotes the strain in front of the pile, $E$ is the elastic modulus of concrete, and $W$ denotes the flexural section modulus of the pile. As seen in Figure 7, the distribution curves of the bending moment of the center pile and secondary side pile under various loading conditions are drawn in relation to the pile top distance. As the bending moment distribution law and value for the middle

and secondary piles are quite similar, they can be considered to have the same stress and deformation laws. As seen in Figure 8a, the bending moment distribution curves of intermediate piles were generated for various loading amounts.

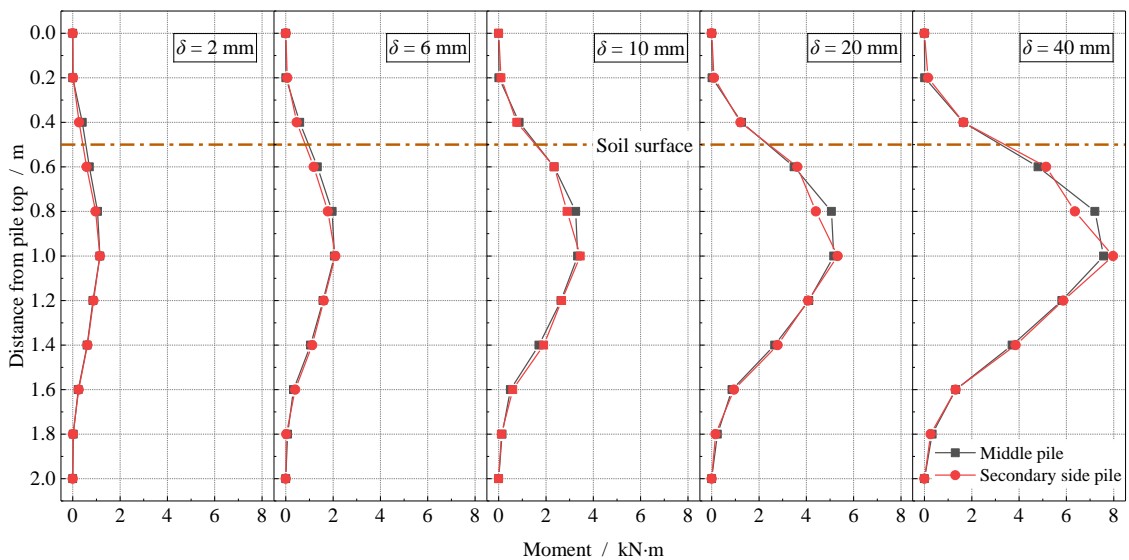

**Figure 7.** Comparison of pile moment between the middle pile and secondary pile.

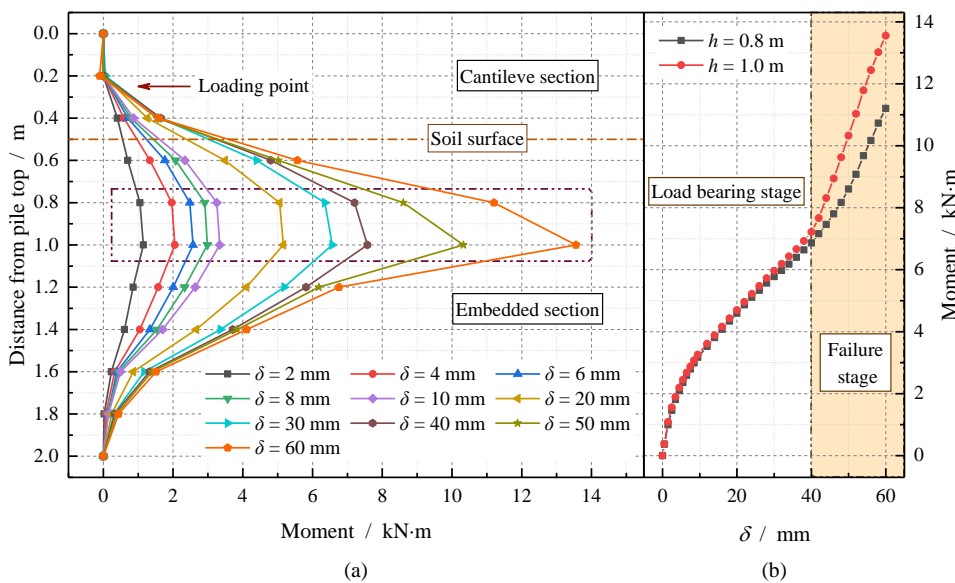

**Figure 8.** Moment of the intermediate pile: (**a**) the bending moment distribution curves of intermediate piles were generated for various loading amounts; (**b**) The curve of the bending moment values at depth of 0.8 m and 1.0 m.

As illustrated in Figure 8a, the pile bending moment is zero above the loading point, whereas the pile bending moment below the loading point has a parabolic distribution, which is consistent with the findings of previous studies [22,25]. The greatest value is between 0.8~1.0 m. When $\delta$ is equal to 40 mm, the maximum value is 7.58 kN·m at 1.0 m from the pile top. As the loading increases, the bending moment values at 0.8 m and 1.0 m rapidly increase. The curve of the bending moment values at these two places is plotted with the loading (Figure 8b), and it is evident that the growth rate of the bending moment increases dramatically at $\delta$ > 40 mm, indicating that the pile is damaged at this point and the failure zone occurs within 0.8~1.0 m. Following the test, it was discovered that noticeable

fissures emerged behind the pile, primarily near 0.9 m from the pile top, or about 1/3 of the anti-slide pile's buried depth.

### 3.3. Analysis of Soil Stress in Front of Pile

Figure 9 is a schematic diagram of the soil in front of the pile. The primary region of investigation is the soil inside the range of piles, which corresponds to the shaded area in the picture. The three measured parts are positioned as follows: Section I-I is centered on the midline between two adjacent piles ($x = 0$), section II-II is centered on the quad-bisect between the two piles ($x = L/4$), and section III-III is centered on the pile side ($x = (L/2 - 2\ \text{cm})$), where $L = 4b$.

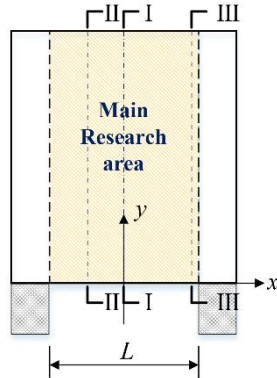

**Figure 9.** Diagram of the model.

The soil stress is analyzed in section I-I. According to the findings of spline curve fitting, the maximum value for $\sigma_x$ should be between 10~20 cm in front of the pile (between two measuring sites), which is referred to as $\sigma_{x,max0}$. Then, fit spline curves for the $\sigma_x$ values at sections II-II and III-III that correspond to distinct loading loads. As illustrated in Figure 10, the coordinate values of the points $\sigma_x = \sigma_{x,max0}$ are searched for arch axis fitting.

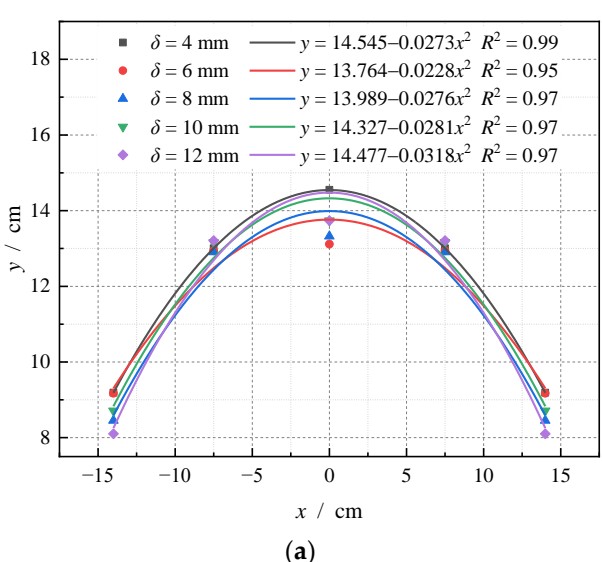

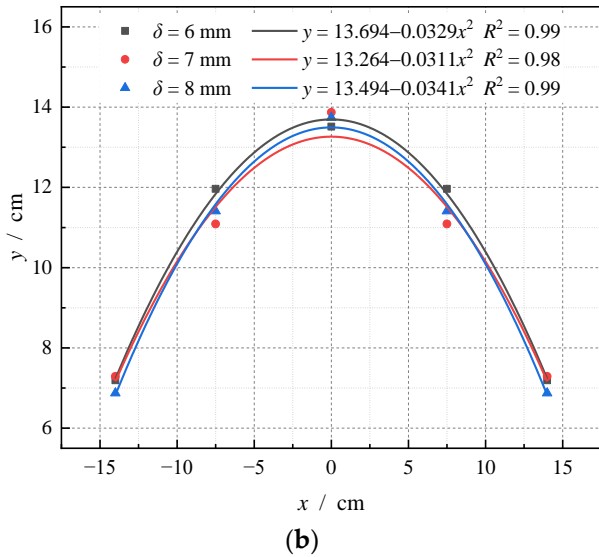

**Figure 10.** Fitting curve of passive earth arch axis in front of piles: (**a**) the earth arch axis at the 15 cm depth; (**b**) the earth arch axis at the 35 cm depth.

As seen in Figure 10a, when the depth is 15 cm and $\delta < 40$ mm, the $\sigma_{x,max0}$ equivalent point of soil in front of the pile can be nicely matched by a parabola. It proves that when the pile spacing is four times the width of the pile, a passive soil arch exists in front of the embedded portion of the anti-slide pile and that its arch axis is a parabola. Except at

$\delta$ = 4 mm, the apex of the soil arch axis advances forward gradually as the loading amount increases, and the arch position approaches $\delta \geq$ 10 mm, showing that the soil arch is totally constructed and that the position and shape of the soil arch do not vary considerably. When $\delta$ > 12 mm, however, the parabolic soil arch cannot be accurately predicted using model test data. At a depth of 35 cm (Figure 10b), evident parabolic arch axes may be obtained when $\delta$ = 6, 7, and 8 mm, but complete arch axes cannot be obtained when $\delta$ > 8 mm, which contradicts previous research findings. At buried depths of 55 cm and 75 cm, no visible arch axis was created.

The following explanations are given for the above failure to match the entire arch axis through observed data once the loading level reaches a specific value: Because $\sigma_{x,max0}$ occurs between two measure points in section I-I, it is difficult to obtain the precise value, and location of $\sigma_{x,max0}$, and so this problem cannot be handled using data processing methods. As a result, it is important to conduct supplemental analysis and rectification of model test findings using an appropriate numerical simulation method.

## 4. Numerical Simulation

### 4.1. Finite Element Model

To supplement the model test results and conduct a systematic analysis of the dynamic evolution of a passive soil arch in front of an embedded piece of an anti-slide pile, ABAQUS was used to create a three-dimensional numerical model for numerical simulation. It should be mentioned that a symmetric finite element model was developed based on past research in order to ensure the finite element grid's calculation correctness and efficiency. The plane of symmetry was established using the middle line of two adjacent anti-slide piles, and only the anti-slide pile and soil model between the two planes of symmetry were established (as shown in Figure 11).

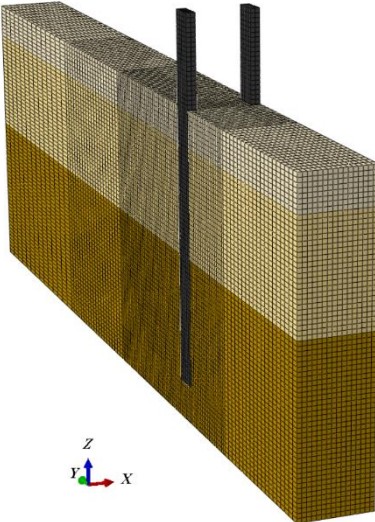

**Figure 11.** The finite element model.

C3D8 element type was used to simulate the finite element model. M-C constitutive model was used to simulate the soil. Table 3 summarizes the parameters. Anti-slide piles were simulated using a linear elastic constitutive model, the parameters of which are listed in Section 2.2.1. The model's bottom boundary is fixed, the vertical direction-direction boundary allows vertical displacement, and the vertical x direction two boundaries are symmetric. The interface between the pile and the earth is the surface-to-surface contact element with a friction coefficient of 0.7. The loading procedure is identical to that used in the model test. y displacement is applied to the anti-slide pile's cantilever portion 25 cm from the pile top. Specific loading steps are detailed in Section 2.4.

The mesh precision of the anti-slide pile and dirt in the x and z axes is 2.5 cm, as seen in Figure 11. The soil close to the anti-slide pile is divided in the y direction with a 2.5 cm grid precision. Beyond 1 m, the mesh precision is steadily raised to 10 cm in front and behind heaps. While the mesh precision required for the primary research area is met, the model's overall mesh count is minimized to maximize computing efficiency.

### 4.2. Accuracy of Numerical Simulation Results

To ascertain the quasi-determination of the numerical simulation results, a comparison is performed between the model's measured data and the numerical simulation results. In section I-I, the bending moment of piles and soil stress are discussed.

#### 4.2.1. Bending Moment of Piles

The comparison curve in Figure 12 illustrates the relationship between the measured data from the model test and the numerical simulation results when different loading loads are drawn. As can be seen, when $\delta \leq 40$ mm, both the observed data and the numerical simulation results follow the same distribution law. The measured values are slightly less than the calculated numbers, and the discrepancy is negligible. On the whole, the numerical simulation findings correspond well with one another. However, when $\delta > 40$ mm, the measured data from model tests gradually exceed the numerical simulation results at 1.0 m and 0.8 m, which is owing to the fact that the numerical simulation of anti-slide piles uses a linear elastic element and does not account the pile's failure stage. The numerical simulation results presented in this article are primarily focused on the $\delta \leq 40$ mm stage.

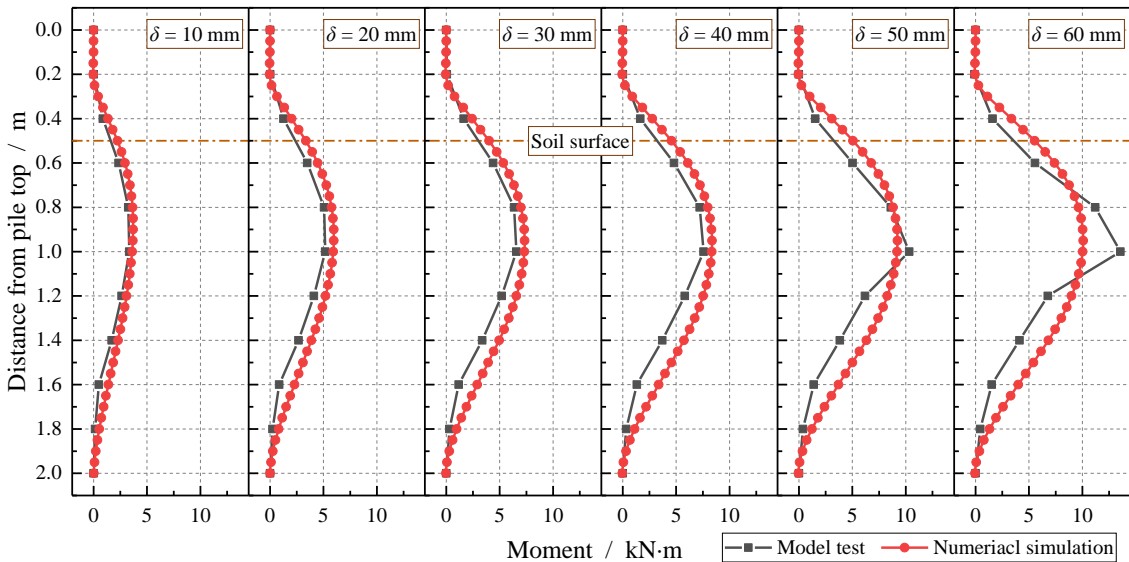

**Figure 12.** Comparison of pile moment between numerical simulation and model test.

#### 4.2.2. Soil Stress in Front of Pile

Passive soil arch in front of a pile is a stress arch generated by nearby piles. Thus, in order to demonstrate the correctness of numerical simulation findings, a comparative examination of soil stress in front of the pile is required. Figure 13 illustrates the comparison curve between measured soil stress model test data and numerical simulations in section I-I. d denotes the buried depth in the figure. As can be shown, both the model test and numerical simulation have a consistent distribution law. When the value is tiny, the two outcomes accord well. As the loading quantity grows, the relative displacement between the pile and the soil increases, resulting in a huge gap between them. The reasons for this are as follows: as the relative displacement between pile and soil increases, the soil in front of the pile experiences noticeable displacement, resulting in a change in sensor position and angle. Additionally, when the sensor's accuracy is considered, a significant

gap exists between the sensor and the numerical simulation. Simultaneously, as the buried depth increases, the consistency between the two values increases, and as the buried depth increases, the relative displacement of pile dirt lags behind the loading. As can be observed from the preceding analysis, the numerical simulation results in this work accurately reflect the variation law of the soil stress field in front of the model test pile, implying that the numerical simulation-based supplemental analysis of the model test is logical and effective.

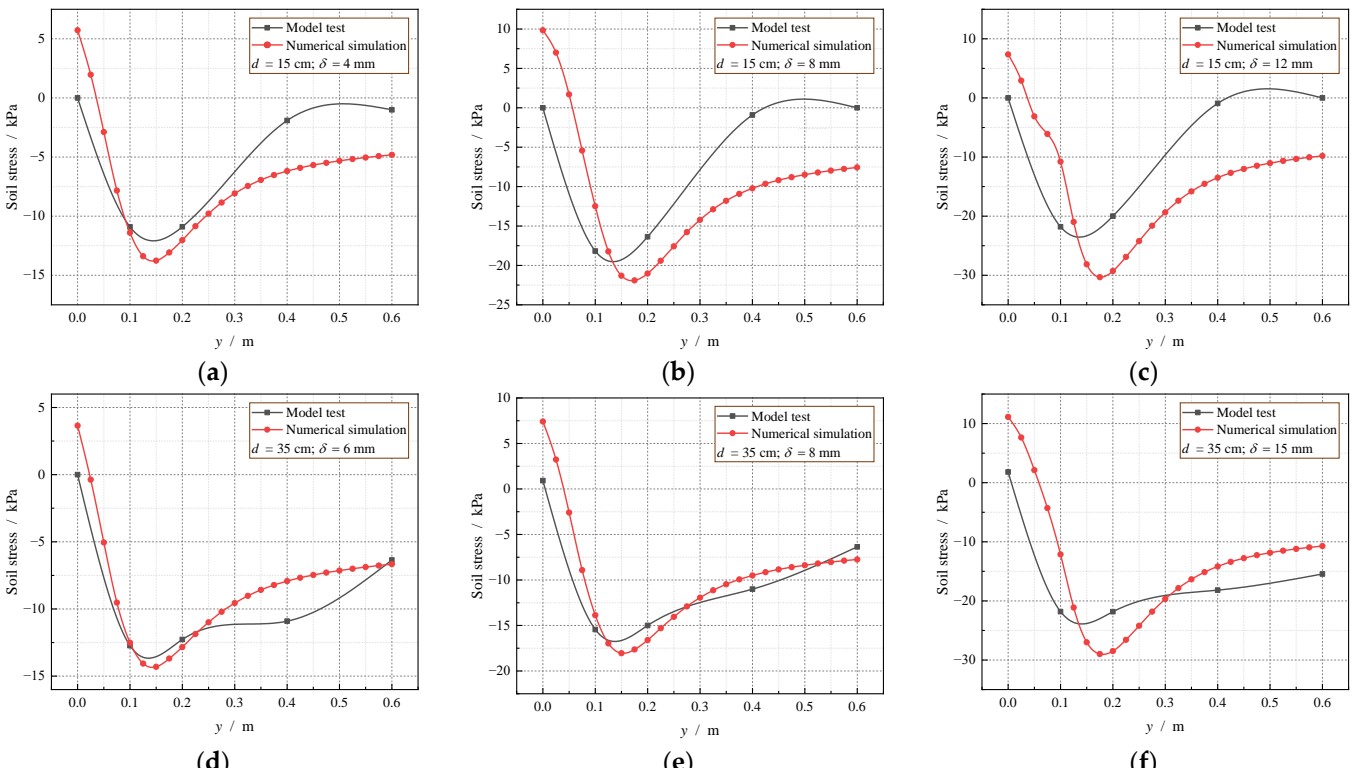

**Figure 13.** Comparison of soil stress between numerical simulation and model test: (**a**) *d* (buried depth) =15 cm, $\delta$ = 4 mm; (**b**) *d* =15 cm, $\delta$ = 8 mm; (**c**) *d* =15 cm, $\delta$ = 12 mm; (**d**) *d* =35 cm, $\delta$ = 6 mm; (**e**) *d* =35 cm, $\delta$ = 8 mm; (**f**) *d* =35 cm, $\delta$ = 15 mm.

As illustrated in Figure 13, $\sigma_{x,max0}$ appears at 0.1~0.2 m in front of the pile and advances in lockstep with the loading amount, exactly between the model test sites on both sides. As a result, when the value is big, a large gap between the model test results and numerical simulation results arises, and the phenomena described in Section 3.3 of the whole arch axis being unable to be fitted through the observed data appears.

### 4.3. Axis and Space Form of Passive Soil Arch in Front of Piles
4.3.1. Axis of Passive Soil Arch in Front of Pile

The passive soil arch axes in front of piles were fitted with varying loading levels at depths of 15 cm, 35 cm, 55 cm, and 75 cm, as illustrated in Figure 14. By examining the variation trend of the arch axis, it is possible to deduce the following:

(1)  The passive soil arch in front of piles is related to the relative displacement between the piles and the soil. With increasing buried depth and the same force condition, the relative displacement of pile-soil reduces. Thus, the shallower the buried depth, the less loading is required, and the sooner the arch is destroyed. Passive soil arches at 15, 35, and 55 cm depth can exhibit four distinct stages as the load increases: formation, development, completion, and destruction.

(2)  Prior to failure, the *y* value of the apex of the passive soil arch axis at a depth of 15 cm is approximately 19.15 cm. At depths of 35 and 55 cm, the passive soil arch appeared to have reached a stable stage, indicating that it had entered the completion phase. As

well, the $y$ values of the apex of the passive soil arch axis were all about 20 cm, with little difference in shape and position, slightly larger than the buried depth of 15 cm. At a buried depth of 75 cm, the passive soil arch varies continually and is not stable, indicating that the passive soil arch is in the development stage.

(3) When the buried depth exceeds 85 cm, no visible passive soil arch is formed as $\delta \leq 40$ mm.

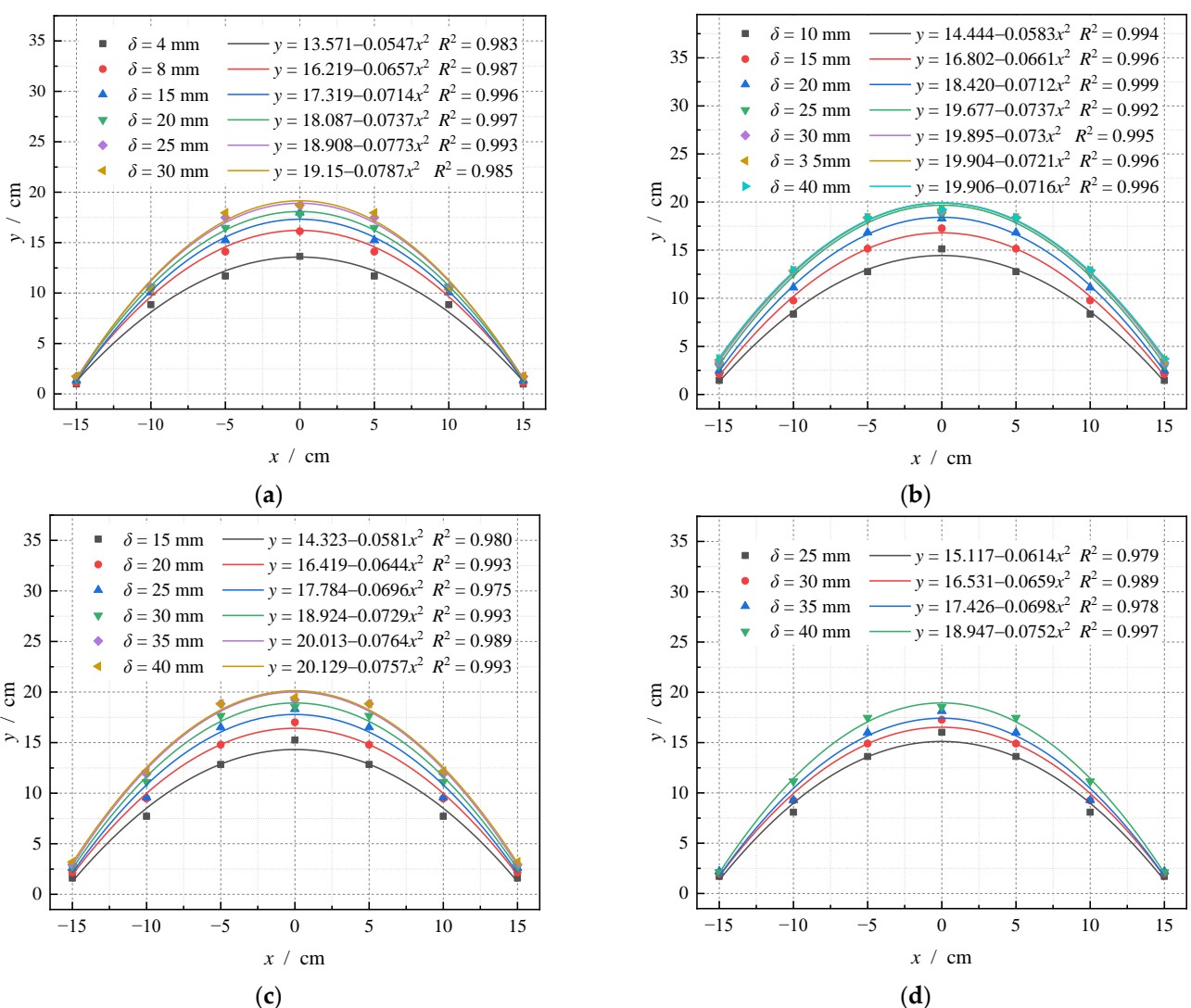

**Figure 14.** Fitting curve of passive earth arch axis in front of piles by numerical simulation: (**a**) the earth arch axis at the 15 cm depth; (**b**) the earth arch axis at the 35 cm depth; (**c**) the earth arch axis at the 55 cm depth; (**d**) the earth arch axis at the 75 cm depth.

### 4.3.2. Spatial Form of Passive Soil Arch

To investigate the spatial distribution variation of the passive soil arch in front of piles as a function of loading amount, a distribution curve of the y value of the soil arch vault with buried depth was produced, as shown in Figure 15. As buried depth increases, a passive soil arch in front of the pile is generated and progressively destroyed. Additionally, the following outcomes can be obtained:

(1) When the same amount of load is applied, the spatial distribution of passive soil arch essentially follows the distribution rule that along the buried depth, gradually approaching the pile front.

(2) When the buried depth is less than 15 cm, the loading amount associated with the passive soil arch from formation to failure is negligible, and its effect on the bearing

capacity of the embedded section of the anti-slide pile is negligible. The passive soil arch is most prevalent in the 15~80 cm buried depth range.

(3) As stress increases, the passive soil arch in front of the pile gradually develops downward along the buried depth. Gradually, the three-dimensional surface grows, and the entire structure slides away from the pile. Additional soil arching from top to bottom till the anti-slide piling structure fails.

To create the three-dimensional surface of passive soil arch space, the maximum range of the passive soil arch space surface was chosen ($\delta = 30$ mm), as illustrated in Figure 16. As can be observed, the spatial surface of the passive soil arch is separated into three distinct zones based on the buried depth: (1) when the buried depth is greater than 45 cm, with the decrease in the buried depth, the soil arch vault and the arch foot gradually move to the loading direction, and the shape of the arch axis changes constantly, which belongs to the development stage. (2) With the further decrease of buried depth, the shape and spatial position of soil arch basically do not change, which is called the completed stage. (3) When the buried depth is less than 25 cm, the passive soil arch enters the completion stage, but its shape is different due to the change of buried depth and soil properties.

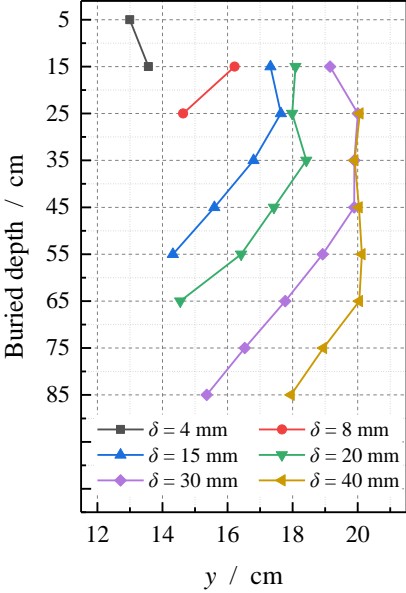

**Figure 15.** Distribution of passive soil arch vault in front of pile with buried depth.

In accordance with the investigation of the three-dimensional surface of the passive soil arch, the displacement of the pile and soil grows gradually as the buried depth decreases, which may be regarded as passive soil arch changes related with pile soil displacement. The law is compatible with Section 4.2's conclusion. It indicates that the complete process of generating a passive soil arch in front of an embedded anti-slide pile may be divided into four stages: formation, development, completion, and destruction within 4 times the pile width of pile spacing.

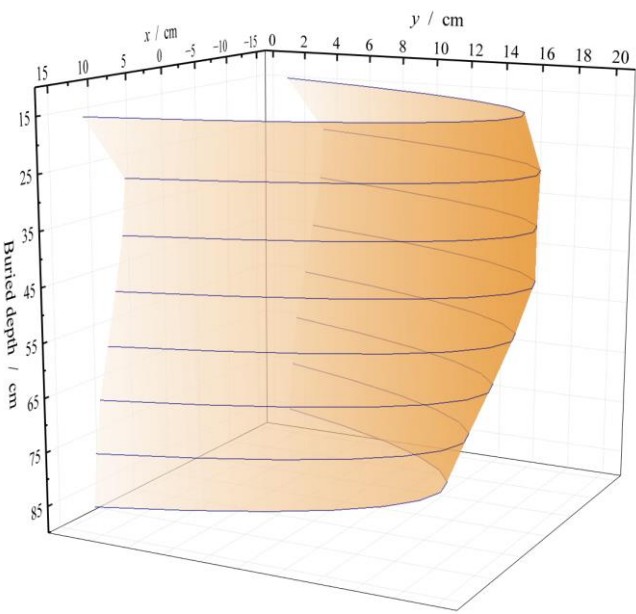

**Figure 16.** Three-dimensional surface of passive soil arch ($\delta$ = 15 mm).

## 5. Conclusions

The existence of a passive soil arch in front of anti-slide piles with a pile spacing of four times the pile width is demonstrated through a model test. The bending strain of the anti-slide pile body is investigated, as well as the shape and development law of the passive soil arch axis in front of the pile. Simultaneously, numerical simulation is utilized to support the model test, and the three-dimensional distribution law of the passive soil arch in front of the pile is investigated. The following are the major conclusions:

(1) By installing side piles, the boundary effect problem can be greatly alleviated. It is proposed that side piles be placed beyond the main focus of the study in order to eliminate boundary effects in the pile row and pile group model tests. The jack's force on the anti-slide pile steadily increases as the load grows, and the growth rate gradually decreases. When the loading amount reaches $\delta \geq 40$ mm, the thrust becomes essentially constant as the loading amount increases, and the anti-slide pile system enters the failure stage.

(2) The parabola distribution of the pile's bending strain after loading is referred to as the parabola distribution. When the strain value exceeds 40 mm, it rapidly increases between 0.8 m and 1.0 m from the pile top. This shows that bending failure occurs on the pile body at this position, with the failure point located between 0.8 and 1.0 m. Follow-up observation confirms that the breakdown point is around 0.9 m from the pile top. Simultaneously, the pile strain values obtained using model test and numerical modeling accord well when $\delta \leq 40$ mm.

(3) The parabola form can be used to fit the $\sigma_{x,max0}$ corresponding points at 15 and 35 cm depth. The existence of a passive soil arch in front of a pile with a spacing of four times the pile width is demonstrated. Simultaneously, when the loading amount is small, the measured value of the model test agrees well with the numerical simulation results, demonstrating the reasonableness of the numerical model presented in this article.

(4) The soil arch evolution law can be divided into four stages within a pile spacing of four times the width of the pile: formation, development, completion, and destruction. With increasing loading, passive soil arches in front of piles are produced and destroyed. As loading increases, the three-dimensional surface formed by the passive soil arch in front of the pile eventually develops downward along the buried depth and moves toward the loading direction in its entirety. Additionally, the soil arch destroys from top to bottom until the anti-slide piling mechanism fails.

Although there are important discoveries revealed by these studies, there is also a limitation. That is, this paper relies on the model test of pile spacing four times pile width for research and analysis, and the working condition is relatively uniform. Subsequently, the passive soil arch effect in front of the pile should be studied under multiple working conditions to further investigate the influencing factors of passive soil arch in front of the pile, and a method for calculating anti-slide pile bearing capacity taking passive soil arch effect into account should be proposed.

**Author Contributions:** Conceptualization, X.R. and L.L.; methodology, X.R. and L.L.; software, X.R. and Y.Z.; validation, X.R., Y.Z. and J.M.; investigation, J.M. and X.A.; resources, J.M. and X.A.; data curation, X.R.; writing—original draft preparation, X.R.; writing—review and editing, X.R., Y.Z. and J.M.; visualization, X.R. and X.A.; supervision, L.L.; project administration, X.R.; funding acquisition, L.L. All authors have read and agreed to the published version of the manuscript.

**Funding:** This work was supported by the National Natural Science Foundation of China (Grant No. 41877285, 41941019), the Fundamental Research Funds for the Central Universities (Grant No. 300102289201, 300102281724) and the Key R & D program of Shaanxi Province (Grant No. 2021SF2-02).

**Data Availability Statement:** Not applicable.

**Conflicts of Interest:** The authors declare no conflict of interest.

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
