# Peer review of "Morphological Evolution of Passive Soil Arch in Front of Horizontal Piles in Three Dimensions"

_buildings, doi:10.3390/buildings12071056_

Round 1

Reviewer 1 Report

Please review the design of the pile and the suitability of this design to the external forces affecting it

Is this research related to reducing the stresses on the pile body as the soil softness increases?

For loose or coarse soil, are these assumptions valid, or will there be a change in the assumptions?

Did you impose in the research the appearance of the gap between the soil and the pile? Explain this in a separate paragraph

Please rewrite the summary more clearly and link it to the results and clarify the longest piles in the research and if they are related to the type and density of soil or not.

Reviewer 2 Report

The manuscript presents an interesting topic in geotechnical engineering. The manuscript is well written and well structured. I would like to accept this paper to be published, subject to minor comments as mentioned below.

1)      Line 105 – Fig.5 should be corrected as Figure 5

2)      Figure 4 is missing in the manuscript

3)      Line 140 – Fig.5 should be corrected as Figure 5

4)      Line 214 – Fig.8 should be corrected as Figure 8

5)      Line 218 – Fig.9 should be corrected as Figure 9

6)      Line 228 – Fig.9 should be corrected as Figure 9

7)      Line 246 – Fig.11 should be corrected as Figure 11

8)      Line 251 – Fig.11 should be corrected as Figure 11

9)      Line 259 – Fig.11 should be corrected as Figure 11

10)   Line 278 – Fig.12 should be corrected as Figure 12

11)   What are the material model parameters used to simulate the pile?

12)    Line 338 – Fig.14 should be corrected as Figure 14

13)   Line 352 – Fig.15 should be corrected as Figure 15

14)   Line 388 – Fig.17 should be corrected as Figure 17

Reviewer 3 Report

This paper presents an investigation about the evolution of passive soil arch which is present in front of horizontal piles considering the three dimensions. In general, the content of the paper is solid. The Authors demonstrate the accuracy of their model by finite element analysis. Results are very promising. I would recommend this paper for publication after the following revisions are addressed by the Authors.

1.      By the end of the Abstract Section, please document the main findings of this paper. I think that in its current form are not well-expressed the principal results.

2.      I am not sure if the way how the Authors are citing is proper. Please revise the guide for Authors to make sure that the citing format is the correct one.

3.      The Introduction Section is good. However, please try to expand a little bit more the discussion about the soil arch effects.

4.      By the end of the Introduction Section, please document the main contribution of this paper to the Buildings Journal. In other words, please justify why this paper deserves to be published.

5.      In Figure 1, please include one more picture illustrating the Model Box in a different angle.

6.      The testing set up illustrated in Figs. 1 and 2, is reported in any guideline or standard? If not, why did the Authors use such a configuration for the testing?

7.      Please include the reference(s) for the data presented in Table 1. In other words, where are the similarity constants presented in Table 1 coming from? Are they just assumed by the Authors? Please justify this.

8.      Please include more details about the reinforcing steel illustrated in Figure 3. How did the authors obtain those longitudinal bars and stirrups?

9.      Why only three soil layers were used for the experiment?

10.   Please explain more in detail the loading protocol presented in Section 2.4.

11.   In Section 3.3, how are the Authors measuring the soil stress? Are the Authors using sensors or an analytical method? I cannot follow this in the manuscript.

12.   In Section 4.1, the Authors are declaring that ABAQUS was used for the numerical simulation. Please justify the use of ABAQUS instead of any other Finite Element software.

13.   I’m sure that the ABAQUS model is very accurate. This is a very powerful Finite Element software. However, I’m a little bit concern about the efficiency (computational time), it seems that the authors used a 2.5 cm mesh for the model. In this sense, how long did the finite element analysis take? Is this time feasible for practical applications of this type of analysis or some assumptions must be done to accelerate the process? Please justify this.

14.   In the Conclusions Section, please include the limitations of this study.

Round 2

Reviewer 1 Report

non